# What is the 'voltage drop' when an effective health promoting intervention for older adults—Choose to Move (Phase 3)—Is implemented at broad scale?

Heather A. McKay[1,2]*, Heather M. Macdonald[1,2], Lindsay Nettlefold[1,2], Katie Weatherson[1], Samantha M. Gray[1,3], Adrian Bauman[4], Karim M. Khan[2,5], Joanie Sims Gould[1,2]

**1** Active Aging Research Team, University of British Columbia, Vancouver, BC, Canada, **2** Department of Family Practice, University of British Columbia, Vancouver, BC, Canada, **3** Aging and Population Health Lab, Department of Biomedical Physiology and Kinesiology, Simon Fraser University, Burnaby, BC, Canada, **4** Sydney School of Public Health, University of Sydney, Sydney, NSW, Australia, **5** School of Kinesiology, Faculty of Education, University of British Columbia, Vancouver, BC, Canada

* heather.mckay@ubc.ca

**Data Availability Statement:** Data cannot be shared with anyone outside of the project team, as consent was not obtained from participants for

## Abstract

### Background

Choose to Move (CTM), an effective health-promoting intervention for older adults, was scaled-up across British Columbia, Canada. Adaptations that enable implementation at scale may lead to 'voltage drop'—diminished positive effects of the intervention. For CTM Phase 3 we assessed: i. implementation; ii. impact on physical activity, mobility, social isolation, loneliness and health-related quality of life (impact outcomes); iii. whether intervention effects were maintained; iv) voltage drop, compared with previous CTM phases.

### Methods

We conducted a type 2 hybrid effectiveness-implementation pre-post study of CTM; older adult participants (n = 1012; mean age 72.9, SD = 6.3 years; 80.6% female) were recruited by community delivery partners. We assessed CTM implementation indicators and impact outcomes via survey at 0 (baseline), 3 (mid-intervention), 6 (end-intervention) and 18 (12-month follow-up) months. We fitted mixed-effects models to describe change in impact outcomes in younger (60–74 years) and older ($\geq$ 75 years) participants. We quantified voltage drop as percent of effect size (change from baseline to 3- and 6-months) retained in Phase 3 compared with Phases 1–2.

### Results

Adaptation did not compromise fidelity of CTM Phase 3 as program components were delivered as intended. PA increased during the first 3 months in younger (+1 days/week) and older (+0.9 days/week) participants (p<0.001), and was maintained at 6- and 18-months. In all participants, social isolation and loneliness decreased during the intervention, but increased during follow-up. Mobility improved during the intervention in younger participants

this. The UBC Clinical Research Ethics Board can be approached with data requests, and they can provide options for sharing study data. The University of British Columbia's Clinical Research Ethics Board can be reached at: pia.ganz@ubc.ca (Pia Ganz, Manager).

**Funding:** JSG and HMM received funding from the Canadian Institutes of Health Research (PJT-153248) for this work (https://cihr-irsc.gc.ca). The British Columbia Ministry of Health also provided funds to support delivery of Choose to Move (http://gov.bc.ca/health/). The funders had no role in study design, data collection and analysis, decision to publish, or preparation of the manuscript.

**Competing interests:** The authors have declared that no competing interests exist.

only. Health-related quality of life according to EQ-5D-5L score did not change significantly in younger or older participants. However, EQ-5D-5L visual analog scale score increased during the intervention in younger participants (p<0.001), and this increase was maintained during follow-up. Across all outcomes, the median difference in effect size, or voltage drop, between Phase 3 and Phases 1–2 was 52.6%. However, declines in social isolation were almost two times greater in Phase 3, compared with Phases 1–2.

## Conclusion

Benefits of health-promoting interventions—like CTM—can be retained when implemented at broad scale. Diminished social isolation in Phase 3 reflects how CTM was adapted to enhance opportunities for older adults to socially connect. Thus, although intervention effects may be reduced at scale-up, voltage drop is not inevitable.

## Introduction

Physical activity is associated with myriad social and physical health benefits, yet most older adults fail to meet physical activity guidelines (150 min/week) [1]; 94% report sitting more than 8 hours/day [2]. While many interventions effectively increased older adults' physical activity [3], fewer than 2% of the 400 unique interventions published in the last three decades were scaled-up to reach a broader population of older adults [4, 5]. Scale-up is defined as "deliberate efforts to increase the impact of successfully tested health innovations so as to benefit more people and to foster policy and programme development on a lasting basis" [6, pg. 9]. We envisage the process of scale-up as a continuum; scale-up may follow typical or atypical pathways from developing the intervention to its dissemination at broader and broader scale [7]. To have population level health impact, physical activity interventions that demonstrated effectiveness in small, controlled settings, must be scaled-up more broadly [8].

To enable widespread implementation, interventions and implementation strategies (i.e., "methods or techniques used to enhance the adoption, implementation, and sustainability of a clinical program or practice" [9, pg. 2]) often need to be adapted [10]. Adaptations may increase as scale up proceeds to reach and engage broader, more diverse populations, to enhance program 'fit' or participants or to address delivery partner needs [11–13]. Adaptations may counter a decrease in participant-level benefits that occur when an intervention is implemented at broader scale–-a phenomenon referred to as a scale-up penalty [13–15] or voltage drop [16, 17]. Effect sizes were, on average, 60% smaller across 10 scaled-up physical activity trials compared with corresponding efficacy trials (pre-scale-up) [14]. Only one of the 10 scaled-up trials (CHAMPS III) targeted older adults; the effect size for physical activity was 69% lower in the scaled-up version of CHAMPS II, compared with the pre-scale trial [18].

In our broad program of implementation research, we address knowledge gaps related to scale-up of health-promoting interventions targeting older adults [5]. Specifically, here we report how we implemented and evaluated a physical activity and social connectedness program for older adults—Choose to Move (CTM Phase 3)—at broad scale. CTM is a flexible, evidence- and choice-based program for older adults in British Columbia (BC), Canada [19–21]. Briefly, CTM incorporates foundational elements of the Community Healthy Activities Model Program for Seniors (CHAMPS) [18, 22, 23] including principles of behaviour change [24, 25] (e.g., goal setting, support, feedback) to help older adults to set physical activity goals, address barriers to physical activity, choose among physical and social activities provided in their

communities, and interact and share challenges and solutions to being active with their peers. CTM Phases 1 and 2 (initial scale-up; referred to as Phases 1–2 throughout this manuscript) were implemented in 2016–17 [19–21]; those phases delivered 56 CTM programs in 22 geographically-distinct communities across BC and reached 534 participants. CTM Phases 1–2 significantly improved physical activity, mobility, and decreased social isolation and loneliness in low active older adult (≥60 years) participants, particularly amongst those aged 60–74 years [20, 26]. Some health benefits were maintained 12 months after the intervention ended [27]. Specifically, among younger participants (60–74 years), intervention-related benefits in mobility, social isolation and loneliness were maintained 12 months after CTM ended, whereas in older participants (≥ 75 years), only decreased loneliness was maintained over 12 months.

From this solid foundation, we incorporated feedback from key stakeholders to systematically adapt CTM and its implementation strategies for CTM Phase 3 for scale-up across more communities in BC (broad scale-up; 2018–2020) [12]. Thus, our CTM Phase 3 intervention study has four objectives:

1. **to describe** implementation indicators associated with delivery of CTM Phase 3 intervention at broad scale;

2. **to evaluate** whether the CTM Phase 3 intervention increased older adults' physical activity (primary outcome), mobility and health-related quality of life (secondary outcomes), and reduced feelings of social isolation and loneliness (secondary outcomes);

3. **to assess** whether health benefits (if any) were maintained 12 months after the CTM Phase 3 intervention ended; and

4. **to assess** the voltage drop (if any) of CTM Phase 3 intervention benefits (compared with Phases 1–2).

## Methods

### Choose to Move

We describe the implementation and scale-up frameworks that guide us [28, 29], design, community partnership development and implementation of CTM at small scale elsewhere [19–21] (ClinicalTrials.gov identifier: NCT05497648; retrospectively registered). We did not prospectively register CTM as a clinical trial because we felt it aligned more closely with the definition of an observational study [30]. Community partners deliver CTM; older adults enrol in the program directly through these organizations and participation in the evaluation is optional. However, we acknowledge the benefits of registering observational trials [31] and we therefore retrospectively registered our study.

At the organization level, CTM is designed to build community capacity to support awareness of, and access to, local health-promoting opportunities. At the individual (participant) level, CTM provides support for older adults to set personalized (based on preference and ability) physical activity goals, and address barriers to physical activity. CTM participants receive social support from fellow participants and their activity coach. We reported the methods for CTM initial scale-up studies elsewhere [20, 27]. This report focuses on CTM Phase 3 (2018–2020) unless we specifically refer to the earlier Phases.

Activity coaches who delivered CTM Phase 3 programs completed self-directed and interactive online training on Moodle™. Training sessions included an overview of CTM intervention components (i.e., one-on-one consultations, group meetings, check-ins) and case studies that illustrated common challenges. During the first 3 months of the 6-month program, activity coaches provided participants: 1) one 60-minute one-on-one consultation, 2) five 60-minute motivational group meetings (up to 15 participants/group; twice in months 1 and 2, once in month 3), and 3) 3 check-ins (15 min, on average; once per month by phone, email or in person). Group meetings took place at local community centres and YMCA facilities. During the last 3 months of the program, coaches provided less support (3 telephone calls). We summarize key differences in the intervention and implementation strategies between CTM Phase 3 and Phases 1–2 in Table 1.

The CTM central support unit (comprised of highly qualified staff from the Active Aging Research Team [32]) worked with delivery partners and coaches to scale-up CTM in eight cycles with staggered start dates between January 2018 and January 2020 (Fig 1). Each cycle included between 4 and 37 programs for a total of 165 programs across cycles 1–8. In Phase 3, forty-seven coaches delivered CTM; each coach delivered a median of 5 CTM programs (range 1–19).

## Study design

To evaluate CTM, we used a type 2 hybrid effectiveness-implementation pre-post study design [33]. Data collection occurred at 0 (baseline), 3 (mid-intervention), 6 (post-intervention) and 18 (12 months post-intervention) months. The University of British Columbia Clinical Research Ethics Board (H15–02522) approved all study procedures.

## Participants

Delivery partner organizations and activity coaches recruited older adults in their community using a variety of strategies (e.g., local promotions such as program guides, posters, and information sessions; media advertisements (including social media); CTM website - https://www.choosetomove.ca/; word of mouth). Eligible participants were community-dwelling males and females aged ≥60 years, English speaking, and low active (self-reported <150 minutes/week of physical activity) with no contra-indications to physical activity participation (Physical Activity Readiness-Questionnaire+ [34], Get Active questionnaire [35] or physician clearance). Participants aged 50 to 59 years were also eligible to participate in extenuating circumstances following recommendations from coaches (e.g., 58-year-old individual who suffered a stroke and would benefit from participating in CTM); however, delivery partner organizations did not actively attempt to recruit participants in this age group.

Programs delivered in the final cycle of Phase 3 (cycle 8, January-February 2020 start date, n = 26 programs) were significantly impacted by the COVID-19 pandemic. We worked with our delivery partners to adapt the program so that CTM could be delivered online/virtually. The 18-month CTM follow-up evaluation for participants in CTM cycles 4–7 was also conducted during the pandemic (June 2020-April 2021).

For our implementation evaluation, all coaches who delivered programs in Phase 3 (cycles 1 through 7; n = 47) were invited to participate. All provided informed consent.

## Data collection–implementation evaluation

To assess implementation of CTM, we invited older adult participants to complete feedback surveys at 3 months (distributed with outcome evaluation questionnaires, described below).

**Table 1. Differences between Choose to Move Phases 1 & 2 and Phase 3.** Reproduced from Gray et al. [12] with permission from Springer Nature.

| Phases 1 & 2 | Phase 3 |
|---|---|
| • Program structure:<br>  ○ No information session prior to program start<br>  ○ Initial consultation during same week as 1st group meeting<br>    ○ 4 group meetings<br>    ○   ○ 10 check-ins<br>    ○ Group meetings:<br>    ○   ○ Health topics covered in group meetings (active travel included in every meeting)<br>    1 Physical activity & chronic conditions<br>    ○   2 Chronic disease self-management<br>    ○   3 Reducing stress & easing anxiety<br>    ○   4 Review CTM principles & behaviour change<br>    5 N/A<br>  ○ No designated movement breaks during meetings<br>  ○ No formal integration of social connectedness<br>Check-ins:<br>  ○ Physical activity action plan and goal setting between activity coach and participant<br>    ○ Extensive section on pain management<br>  ○ No questions about future goals or physical activity plans<br>  ○ No motivational interviewing prompts<br>  ○ No focus on the CTM as a social 'community' of support<br>Activity coach training:<br>  ○ Qualification–certified fitness leaders or kinesiologists<br>  ○ Activity coaches hired by delivery partners<br>  ○ Training specific to fitness professionals (as described above)<br>  ○ Training delivered as day-long, in-person session and a hardcopy manual<br>  ○ Social connectedness not formally integrated into the activity coach training<br>• Program operations:<br>  ○ Lead time for program delivery not standardized across delivery sites (often < 3 months)<br>  ○ No formal communication plans between delivery partners and delivery sites<br>  ○ No central recruitment resource available to delivery sites<br>  ○ Delivery site agreements vague; exact roles and responsibilities unclear<br>  ○ Promotion and recruitment materials lacked emphasis on benefits of CTM to participants | • Program structure:<br>  ○ Information session 2 weeks prior to program start<br>  ○ Initial consultation one week prior to 1st group meeting<br>    ○ 5 group meetings<br>    ○ 6 check-ins<br>    ○ Group meetings:<br>    ○ Health topics covered in group meetings (active travel only in 1st meeting as 'incidental activity')<br>1. Physical activity & social connection<br>2. Healthy weight management & nutrition<br>3. Stress & anxiety<br>4. Brain health & preventing injury<br>5. Revisit your goals & celebrate!<br>    ○ Prescribed movement breaks during group meetings<br>    ○ Group meeting slides prescriptive for group and paired discussions; contact information formally included in each CTM participant group<br>• Check-ins:<br>    ○ Physical activity action plan and goal setting between activity coach and participant<br>    ○ Pain section reduced<br>    ○ Increased emphasis on compliance to action plan (revisit goals, set new goals, future plans)<br>    ○ Added prompts for motivational interviewing<br>    ○ Focus on the CTM as a social 'community' of support<br>• Activity coach training:<br>    ○ Qualification–anyone with experience in fitness leadership or with older adults<br>    ○ Activity coaches hired by delivery partners in consultation with recreation coordinators<br>    ○ Training expanded to encompass knowledge base of activity coaches who may not be fitness professionals<br>    ○ Training delivered in self-directed online platform and interactive practical component<br>    ○ Training adapted to emphasize social connections; group meeting presentation slides more prescriptive for group and pair discussion<br>• Program operations:<br>    ○ Every delivery site given 3–6 months lead time prior to program start-up<br>    ○ Each delivery site given site-specific communication plans, which includes: implementation and site activity checklists, as well as regular communication plan<br>    ○ Central recruitment resource available (www. choosetomove.info)<br>    ○ Agreements modified to more clearly articulate expectations regarding promotion and recruitment<br>    ○ Promotion and recruitment materials modified to highlight the benefits of CTM to participants |

Activity Coaches completed a program delivery survey at 3 and 6 months and a participant engagement survey at 6 months. We collected survey data from coaches using REDCap (Research Electronic Data Capture) [36] electronic data capture tools hosted at the University of British Columbia. We emailed survey links to coaches and responses were entered directly into our REDCap database. For the participant engagement surveys, we used the Email Helper application within REDCap and Python (https://realpython.com/) to send coaches unique REDCap links for each participant in their group.

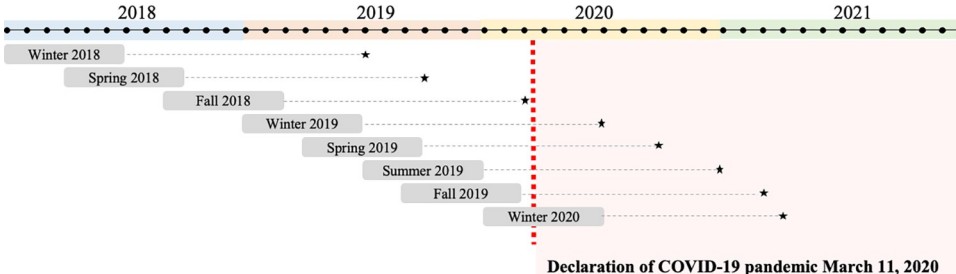

\* Approximate timing of the post-intervention follow-up. For cycles 1 (Winter 2018) through 7 (Fall 2019), follow-up measures were collected 12 months after the intervention ended. For cycle 8 (Winter 2020), follow-up measures were collected 9 months after the intervention ended.

**Fig 1. Choose to Move Phase 3 Timeline.** Overview of the timing of Choose to Move Phase 3 programs delivered between January 2018 and January 2020.

## Implementation indicators

We assessed CTM dose delivered and fidelity as implementation indicators [37]. To assess *dose delivered* (i.e., intended units of each intervention component delivered to participants by the delivery team [38]), we report the number of group meetings and telephone check-ins delivered by coaches. To assess *fidelity* (i.e., the extent to which the intervention was implemented as it was prescribed in the intervention protocol by the delivery team [39]), we describe the extent to which coaches used and applied intervention materials (e.g., group meeting slides) and core functions (e.g., goal setting, action planning).

We determined *dose received* (i.e., the extent to which participants engaged or interacted with, are receptive to, or used the intervention [40]) as the number of motivational group meetings and telephone check-ins participants attended (from participant engagement survey). In addition, we describe *participant responsiveness* (i.e., "the degree to which the program stimulates the interest or holds the attention of participants" [29, pg. 329]) as participant satisfaction with CTM (after the first three months) and coaches' perception of how interactive, enthusiastic, interested and engaged participants were during group meetings in the first three months of the intervention).

## Data collection–impact evaluation

We replicated measurement protocols described in our Phases 1–2 CTM impact evaluation [20, 27]. At baseline, 3, 6 and 18 months, we collected survey-based data from all participants using a paper questionnaire. Participants completed their surveys at group meetings (baseline, 3 months) or at home (6 and 18 months) via mailed surveys; participants who required extra assistance (<1%) at 6 and 18 months completed their survey over the phone with a trained research assistant. For participants in cycle 8 (Winter 2020), post-intervention follow-up data collection occurred 9 months after the intervention ended instead of 12 months as for the other 7 cycles. We made this change to allow for cycle 8 follow-up data to be used as control data for another study being conducted by our research team during the pandemic (NCT04592614).

## Demographic characteristics

At baseline only, we asked participants to complete a demographic survey to determine age (years), sex (male, female), self-reported height (cm) and weight (kg) (with which we calculated body mass index (BMI, kg/m$^2$)), educational attainment (secondary school or less, at

least some trade/technical school or college, at least some university), number of chronic conditions (0, 1, ≥2) and self-rated health (very poor, poor, fair, good, excellent; single question [41, 42]). Participants also self-reported their ethnic origin (as per Canadian census response options for visible minorities [43]); we then categorised participants as Asian, white or other/mixed ethnicity.

## Physical activity and mobility

At each measurement time point, participants completed the single-item physical activity measure [44, 45], and self-reported their physical activity as the number of days in the previous week that they accumulated 30 minutes or more of moderate-to-vigorous physical activity. We recently reported that the single-item measure had acceptable convergent validity and was responsive to change, compared with the 41-item CHAMPS questionnaire [46]. Participants also self-reported their capacity for mobility at each time point; we dichotomized responses as either NO difficulty or ANY difficulty walking 400m and/or climbing one flight of stairs [47].

## Social isolation and loneliness

At each measurement time point, we assessed social isolation using a three-item questionnaire [48] adapted from two questions on social contact frequency [49] (social isolation score; range 0–15, where higher scores indicate less social isolation). We assessed loneliness using the UCLA Loneliness Scale (UCLA-3) questionnaire (loneliness score; range 3–9, where higher scores indicate greater loneliness) that shows good internal consistency, discriminant validity, and convergent validity [50].

## Health-related quality of life

At each measurement time point, we assessed health-related quality of life using the EQ-5D-5L [51]. The EQ-5D-5L consists of five dimensions (mobility, self-care, usual activities, pain/discomfort and anxiety/depression) and a visual analogue scale (VAS). Participants were asked to indicate their level of functioning (from 1 "no problems" to 5 "extreme problems") on each of the five dimensions of the EQ-5D-5L. The EQ-5D-5 L describes 3125 distinct health states, with 11111 representing the best and 55555 the worst possible health states. We applied the Canadian EQ-5D-5 L scoring algorithm [52] to generate index scores, which ranged from −0.148 for the worst (55555) to 0.949 for the best (11111) health states. For the VAS, participants were asked to rate their subjective health perception on the 20 cm VAS with endpoints labeled "the worst health imaginable" (0) and "the best health imaginable" (100).

## Statistical analysis

**Sample size.** We assumed 70% recruitment into the CTM Phase 3 evaluation and 15% attrition across the 6-month intervention. We calculated power using the mean change in physical activity (by the single-item measure) observed in the whole cohort in Phases 1–2 ($1.4 \pm 2.1$ days/week). With alpha = 0.05 (one-tailed) and an estimated sample of 963 participants (1620*70% recruitment*15% attrition), we would have >95% power to detect a meaningful change in physical activity of 1 day/week between baseline and 6 months and >90% power to detect a change of 0.5 day/week.

We performed all analyses using Stata (version 13.1; StataCorp, College Station, TX). As CTM programs in cycle 8 (Winter 2020, 26 programs) were not delivered as planned, we excluded participants (n = 203) in cycle 8 programs from our primary analysis. We therefore first examined whether participants who were lost to follow-up (i.e., withdrew from the

program or did not complete the evaluation) or were part of cycle 8 (affected by COVID-19) differed from participants who completed CTM. We used two-tailed chi-squared or Fisher's exact test for categorical variables (sex, age category, ethnicity, education, chronic conditions, mobility limitations, and subset participation) and analysis of variance for continuous variables (body mass index and impact variables). Next, we used two-sided t-tests (for continuous variables) and chi-squared tests (for categorical variables) to compare socio-demographic characteristics at baseline between age groups (60–74 years, $\geq$ 75 years) for those participants in cycles 1–7.

To address our first objective (implementation evaluation), we assessed: 1) age- and sex-related differences in program delivery using Wilcoxon rank sum tests and 2) the relationship between program dose received and impact outcomes using Spearman rank correlations.

To address objectives 2 and 3 (impact evaluation; follow-up), we fitted linear mixed effects models for each continuous impact variable [physical activity (primary outcome), social isolation, loneliness, health-related quality of life (secondary outcomes)] with time (0, 3, 6, and 18 months) as a categorical predictor. We first fit an empty means random intercept model and tested whether random slopes improved model fit using likelihood ratio tests. In model 1, we included sex and age category (60–74 years, $\geq$ 75 years) as fixed effects. Model 2 included additional covariates: delivery partner, program cycle (1–7), baseline mobility limitation (yes/no), number of chronic conditions (0, 1, $\geq$2), education and BMI. In both models, we added fixed effects sequentially and tested interactions with time after the addition of each fixed effect. With the exception of an age × time interaction, the interactions were retained in the model only if the likelihood ratio test was significant ($p < 0.05$). We assessed model fit graphically using residual plots; plots indicated acceptable model fit. Adjusted values were calculated at each time point using the margins command in Stata with a Bonferroni adjustment to account for multiple comparisons between and within age groups. We also used chi-squared tests to assess differences in the proportion of participants with mobility limitations over time (0–3, 0–6, 6–18 and 0–18 months; secondary outcome) within each age group. We used a Bonferroni adjustment to account for multiple comparisons (significance at: 0.05/4 = 0.013). We used a per-protocol approach as participants who withdrew from the program also withdrew from the evaluation. As a supplementary exploratory analysis, we fit the same models with data for participants in cycle 8 to examine the potential impact of COVID-19 on our outcomes. Due to the change in timing of the follow-up measurements for this cohort, we only present the baseline, 3- and 6-month data.

To address our fourth objective (voltage drop with scale-up), we used the approach reported by McCrabb et al. [14] for uncontrolled pre-post studies. We first calculated the effect size for each outcome (with the exception of health-related quality of life and VAS, which were not assessed in Phases 1–2) as the change from baseline. We then calculated the voltage drop (i.e., percent of the effect size reported in CTM Phases 1–2 that was retained in Phase 3) as: (Phase 3 effect size / Phases 1–2 effect size)*100 [20]. Values greater than 100% indicate a greater benefit of the intervention in Phase 3 broad scale-up as compared with Phases 1–2; values of 50% indicate Phase 3 was half as effective as in Phases 1–2; and values less than 0% (negative values) indicate that the direction of the intervention effect in Phase 3 was opposite of the direction in Phases 1–2 [20].

## Results

Of 1668 participants across 165 programs delivered between January 2018 and June 2020 in 39 BC communities, 1216 older adults (72.9%) consented to participate in the evaluation (Fig 2). Of the 1012 participants from cycles 1–7 who consented to participate in the CTM evaluation,

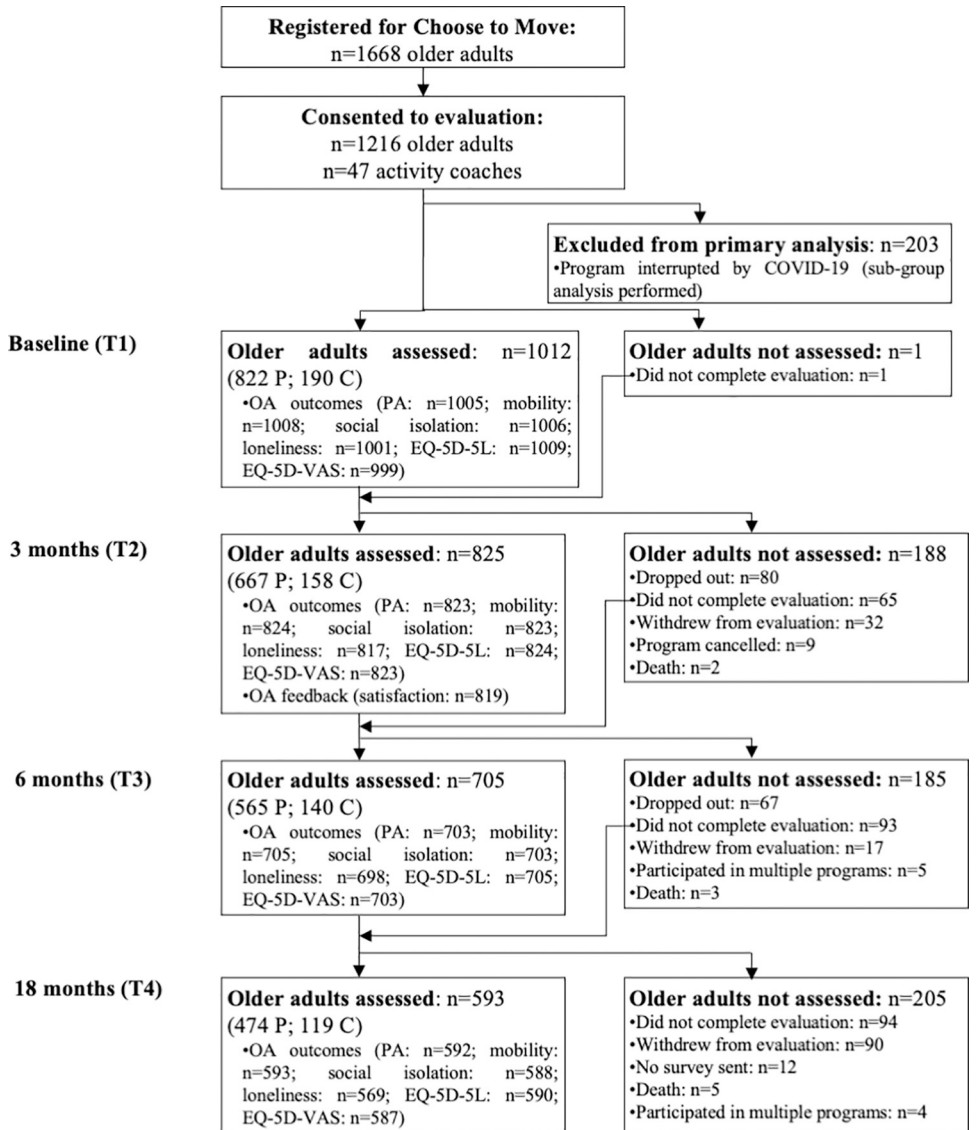

**Fig 2. Participant flow diagram.** Flow of participants through the Choose to Move Phase 3 study.

32.2% (n = 326) were lost to follow-up or withdrew from CTM or the evaluation (Fig 2). For analysis, we included all available data across 4 time points; the number of participants with data at 4, 3, 2 and 1 time points was 531 (52.5%), 190 (18.8%), 149 (14.7%) and 143 (14.1%), respectively. No adverse events were reported by participants. We present baseline characteristics of the CTM Phase 3 cohort in Table 2. Seventy one percent of participants were females of white ethnicity between the ages of 54 and 74 years. Baseline characteristics were similar between participants who completed CTM, those lost to follow-up or those in cycle 8—with the exception of ethnicity (i.e., more participants who completed CTM were white as compared with those lost to follow-up) and education (i.e., fewer participants in cycle 8 had attained a secondary school level of education or less as compared with participants in cycles 1–7 who completed the program) (S1 Table). Only baseline VAS score differed across groups; scores were higher among participants who completed CTM as compared with those lost to follow-up, and those in cycle 8.

**Table 2. Baseline participant socio-demographic characteristics by age group in the Phase 3 Choose to Move cohort (cycles 1–7).** Values are n (%) or mean (standard deviation).

| | Total | <75 years (n = 650) | ≥ 75 years (n = 362) | p-value[a] |
|---|---|---|---|---|
| Participants, n (men / women) | 1012 (196/816) | 650 (123/527) | 362 (73/289) | 0.632 |
| % (men) | 19.4% | 18.9% | 20.2% | |
| Age, mean (SD) | 72.9 (6.3) | 69.0 (3.4) | 79.9 (3.9) | |
| Age category | | | | |
| 54–74 years | 650 (64.2%) | | | |
| ≥75 years | 362 (35.8%) | | | |
| Delivery partner, n (BCRPA / YMCA) | 631 / 381 | 388 / 262 | 243 / 119 | 0.019 |
| BMI, kg/m$^2$ | | | | |
| Men (n = 194) | 29.6 (5.5) | 30.3 (6.2) | 28.2 (3.7) | 0.009 |
| Women (n = 778) | 29.3 (6.5) | 30.4 (6.8) | 27.6 (5.6) | <0.001 |
| Ethnicity, n (%) | | | | |
| White | 892 (88.1%) | 568 (87.4%) | 324 (89.5%) | |
| Asian | 66 (6.5%) | 43 (6.6%) | 23 (6.4%) | |
| Other | 54 (5.3%) | 39 (6.0%) | 15 (4.1%) | 0.439 |
| Educational attainment, n (%) [N = 1009][b] | | | | |
| Secondary or less | 281 (27.8%) | 141 (21.7%) | 140 (38.9%) | |
| Some trade, technical school or college | 356 (35.3%) | 261 (40.2%) | 95 (26.3%) | |
| Some university | 372 (36.9%) | 247 (38.1%) | 125 (34.7%) | <0.001 |
| Chronic Conditions, n (%) | | | | |
| 0 | 150 (14.8%) | 99 (15.2%) | 51 (14.1%) | |
| 1 | 372 (36.7%) | 242 (37.2%) | 130 (35.9%) | |
| ≥ 2 | 490 (48.4%) | 309 (47.5%) | 181 (50.0%) | 0.739 |
| Mental Health Conditions, n (%) [N = 868] | | | | |
| 0 | 652 (75) | 395 (71) | 257 (82) | |
| 1 | 135 (16) | 97 (17) | 38 (12) | |
| ≥ 2 | 81 (9) | 63 (11) | 18 (6) | 0.001 |
| Mobility limitations (walk and/or stair), n (%) [N = 1008] [b] | | | | |
| Yes | 459 (45.5%) | 281 (43.4%) | 178 (49.4%) | |
| No | 549 (54.5%) | 367 (56.6%) | 182 (50.6%) | 0.063 |
| Self-rated health, n (%) [N = 1008] [b] | | | | |
| Very poor, poor or fair for age | 461 (45.7%) | 346 (53.3%) | 115 (32.0%) | |
| Good or excellent for age | 547 (54.2%) | 303 (46.7%) | 244 (68.0%) | <0.001 |

BCRPA: British Columbia Parks and Recreation Association; YMCA: Young Men's Christian Association

[a] *p* values were calculated using two-sided t-tests for continuous variables and chi-squared tests for categorical variables.

[b] Sample size reduced due to missing data for questions regarding level of education, mobility limitations and self-rated health.

## Implementation evaluation–objective 1

*Dose delivered*: Across the 141 programs in cycles 1–7, the 45 trained coaches delivered 705 (100%) group meetings and 4682 (77.1%) check-ins.

*Fidelity*: Activity Coaches provided opportunities for participants to interact with each other during all 5 group meetings in 93.3% of programs; led participants through stretches or movement breaks at 3 or more group meetings in 97.8% of programs; and covered the recommended topic for each group meeting in 97.8–100% of programs. For participant check-ins, 96.2% of coaches reported using motivational interview techniques during most or all of the check-ins; 66.7% of coaches reported helping all of their participants to revisit their goals or set

new goals and review their action plan during the final check-in. Finally, 98.7% of coaches reported using and following CTM resources most or all of the time.

*Dose received*: Of the 1012 participants in cycles 1–7, 1004 (99.2%) attended at least one group meeting or check-in. Of those who did not drop out (n = 705), 76.5% attended ≥80% of the group meetings and 79.2% attended ≥80% of the check-ins. Total dose received (check-ins plus group meetings) did not differ between males and females or between age groups.

*Relationship between dose received and impact variables*: Among impact variables, only social isolation was significantly associated with program dose received at 3 months ($r_s$ = 0.081, p = 0.012). This relationship was apparent in older ($r_s$ = 0.126, p = 0.021), but not younger, participants ($r_s$ = 0.061, p = 0.134).

*Participant responsiveness*: After the first 3 months of the intervention (period during which all group meetings were held), 90.4% of participants were satisfied with CTM. From the coaches' perspective, 77.0% of participants were very or extremely interactive at the group meetings, and 97.8% of participants were enthusiastic, interested and engaged with the CTM content and each other at 3 or more of the group meetings.

## Impact evaluation and follow-up–objectives 2 and 3

We present results of our impact evaluation in Tables 3 and 4. Results were similar for minimally and fully adjusted models; below we focus on the fully adjusted models for younger (54–74 years) and older (≥75 years) participants. For each impact outcome, we present the results for the intervention period first, and results for the follow-up period second. We present results of our exploratory analysis of participant data from cycle 8 (COVID-19) in S2 Table.

*Physical activity*: Among younger participants, physical activity increased from baseline to 3 months (+1.0 days/week; 95% CI: 0.4, 1.5). Gains in physical activity were maintained during the final 3 months of the program; physical activity at 6 months remained significantly higher than at baseline (+0.6; 95% CI: 0.04, 1.2). Among older participants, physical activity increased from baseline to 3 months (+0.9; 95% CI: 0.2, 1.6). Physical activity did not differ between baseline and 6 months.

Among younger participants, physical activity at 18 months (12 months after the CTM intervention ended) did not differ from values at 6 months, but remained significantly higher than at baseline (+0.6; 95% CI: 0.2, 1.6). For older participants, physical activity at 18 months did not differ from values at 6 months, or at baseline.

*Mobility*: Among younger participants, prevalence of mobility limitations decreased by 7.2% from baseline to 3 months. Prevalence of mobility limitations at 6 months did not differ from baseline. Among older participants, prevalence of mobility limitations did not differ between baseline and 3 or 6 months.

At the 18-month follow-up, prevalence of mobility limitations did not differ from baseline in either the younger or older participants.

*Social isolation*: Among younger participants, social isolation score increased from baseline to 3 months (+0.8; 95% CI: 0.5, 1.1; indicating decreased feelings of social isolation) and remained higher at 6 months as compared with baseline (+0.5; 95% CI: 0.1, 0.8). Among older participants, social isolation score increased during the first 3 months of the intervention (+0.5; 95% CI: 0.1, 1.0), but was not significantly different between baseline and 6 months.

In both the younger and older groups, social isolation score decreased during the follow-up period such that values at 18 months were significantly lower than at 6 months and baseline (<75 years: -1.6; 95% CI: -2.0, -1.2; ≥75 years: -2.5; 95% CI: -3.1, -2.0).

*Loneliness*: Among younger participants, loneliness score decreased from baseline to 3 months (-0.2; 95% CI: -0.4, -0.1; reduced feelings of loneliness) and remained lower at 6

**Table 3. Adjusted means (95% confidence interval) for impact outcome measures by time point and age group in the Choose to Move Phase 3 cohort.**

| | Months | Sample size: Full sample / <75 years / > 75 years | Full sample | <75 years | ≥ 75 years | p-value[a] Full sample 0–3 mos 0–6 mos 0–18 mos 6–18 mos | p-value[a] <75 yrs 0–3 mos 0–6 mos 0–18 mos 6–18 mos | p-value[a] ≥75 yrs 0–3 mos 0–6 mos 0–18 mos 6–18 mos |
|---|---|---|---|---|---|---|---|---|
| **Physical activity (# d/wk>30 min)** | 0 | 1005 / 647 / 358 | 2.5 (2.3, 2.6) | 2.4 (2.3, 2.6) | 2.5 (2.3, 2.7) | | | |
| | 3 | 823 / 536 / 287 | 3.4 (3.1, 3.7) | 3.4 (3.0, 3.8) | 3.4 (2.8, 3.9) | <0.001 | <0.001 | 0.008 |
| | 6 | 703 / 453 / 250 | 2.9 (2.6, 3.3) | 3.0 (2.6, 3.5) | 2.8 (2.2, 3.4) | 0.026 | 0.029 | >0.99 |
| | 18 | 592 / 385 / 207 | 2.9 (2.5, 3.2) | 3.0 (2.6, 3.5) | 2.6 (2.0, 3.2) | 0.131 >0.99 | 0.046 >0.99 | >0.99 >0.99 |
| **Mobility [n, (% reporting any limitation)]** | 0 | 1008 / 648 / 360 | 459 (45.5%) | 281 (43.4%) | 178 (49.4%) | | | |
| | 3 | 824 / 536 / 288 | 327 (39.7%) | 194 (36.2%) | 133 (46.2%) | 0.012 | 0.012 | 0.409 |
| | 6 | 705 / 454 / 251 | 286 (40.6%) | 164 (36.1%) | 122 (49.6%) | 0.041 | 0.016 | 0.838 |
| | 18 | 593 / 385 / 208 | 256 (43.2%) | 147 (38.2%) | 109 (52.4%) | 0.358 0.344 | 0.102 0.538 | 0.497 0.418 |
| **Social Isolation (score, 0–15)** | 0 | 1006 / 646 / 360 | 11.0 (10.8, 11.2) | 10.9 (10.7, 11.1) | 11.1 (10.8, 11.4) | | | |
| | 3 | 823 / 536 / 287 | 11.7 (11.5, 11.9) | 11.7 (11.4, 11.9) | 11.6 (11.3, 12.0) | <0.001 | <0.001 | 0.008 |
| | 6 | 703 / 454 / 249 | 11.4 (11.2, 11.6) | 11.4 (11.1, 11.6) | 11.5 (11.1, 11.8) | <0.001 | <0.001 | 0.290 |
| | 18 | 588 / 382 / 206 | 9.5 (9.2, 9.7) | 9.7 (9.5, 10.0) | 8.9 (8.5, 9.3) | <0.001 <0.001 | 0.001 <0.001 | <0.001 <0.001 |
| **Loneliness (score, 3–9)** | 0 | 1001 / 646 / 355 | 4.5 (4.4, 4.6) | 4.7 (4.5, 4.8) | 4.2 (4.0, 4.4) | | | |
| | 3 | 817 / 532 / 285 | 4.3 (4.2, 4.4) | 4.5 (4.3, 4.6) | 4.1 (3.9, 4.3) | <0.001 | <0.001 | 0.754 |
| | 6 | 698 / 451 / 247 | 4.3 (4.2, 4.4) | 4.4 (4.3, 4.5) | 4.0 (3.8, 4.2) | <0.001 | <0.001 | 0.133 |
| | 18 | 569 / 375 / 194 | 4.7 (4.6, 4.9) | 4.8 (4.6, 4.9) | 4.6 (4.4, 4.8) | 0.003 <0.001 | 0.765 <0.001 | 0.001 <0.001 |
| **Health status (EQ-5D-5L)** | 0 | 1007 / 646 / 361 | 0.793 (0.786, 0.800) | 0.785 (0.776, 0.795) | 0.807 (0.794, 0.820) | | | |
| | 3 | 823 / 536 / 287 | 0.800 (0.782, 0.817) | 0.800 (0.778, 0.822) | 0.799 (0.770, 0.828) | >0.99 | >0.99 | >0.99 |
| | 6 | 703 / 452 / 251 | 0.806 (0.788, 0.825) | 0.810 (0.786, 0.833) | 0.800 (0.770, 0.830) | 0.995 | 0.237 | >0.99 |
| | 18 | 587 / 383 / 204 | 0.778 (0.759, 0.800) | 0.777 (0.753, 0.802) | 0.780 (0.748, 0.817) | 0.809 0.074 | >0.99 0.121 | 0.631 >0.99 |
| **VAS (EQ-5D-5L)** | 0 | 999 / 644 / 355 | 69.8 (68.8, 70.9) | 67.2 (65.9, 68.5) | 74.4 (72.7, 76.2) | | | |
| | 3 | 823 / 536 / 287 | 74.6 (72.4, 76.8) | 74.1 (71.3, 76.9) | 75.5 (71.9, 79.1) | <0.001 | <0.001 | >0.99 |
| | 6 | 703 / 453 / 250 | 74.7 (72.4, 77.0) | 73.8 (71.4, 77.1) | 76.4 (72.6, 80.2) | <0.001 | <0.001 | >0.99 |
| | 18 | 587 / 383 / 204 | 72.9 (70.5, 75.3) | 72.4 (69.4, 75.4) | 74.5 (70.4, 78.5) | 0.064 >0.99 | 0.009 >0.99 | >0.99 >0.99 |

d/wk: days/week; Social isolation: higher score indicates a larger social network; Loneliness: lower score indicates lower feelings of loneliness; EQ-5D-5L Health Status and Visual Analog Scale (VAS): higher score indicates better health status

[a] p values for continuous variables were calculated using a fitted linear mixed effects models for each continuous impact variable with time (0, 3, 6, and 18 months) as a categorical predictor. Fixed effects were sex and age category (60–74 years, ≥ 75 years); additional covariates were delivery partner, program cycle (1–7), baseline mobility limitation (yes/no), number of chronic conditions (0, 1, ≥2), education and body mass index. Adjusted values were calculated at each time point using the margins command in Stata with a Bonferroni adjustment to account for multiple comparisons between and within age groups. p values for mobility limitations were calculated using chi-squared tests with Bonferroni adjustment to account for multiple comparisons between time points (significance at: 0.05/4 = 0.013).

**Table 4. Comparison of intervention effect sizes (95% confidence interval) for primary outcomes between Choose to Move Phases 1 & 2 and Phase 3.**

| | Month | Phases 1 & 2 Effect size[a] | | | Phase 3 Effect size [a] | | | Proportion (%) of the efficacy trial effect size achieved with scale-up | | |
|---|---|---|---|---|---|---|---|---|---|---|
| | | Full cohort | <75 yrs | ≥75 yrs | Full cohort | <75 yrs | ≥75 yrs | Full cohort | <75 yrs | ≥75 yrs |
| **Physical activity (# d/wk>30 min)** | 3 | 1.4 (1.2, 1.7) | 1.6 (1.3, 1.9) | 1.0 (0.5, 1.4) | 0.9 (0.5, 1.4) | 1.0 (0.4, 1.5) | 0.9 (0.2, 1.6) | 63.4 | 62.5 | 100 |
| | 6 | 1.1 (0.8, 1.3) | 1.4 (1.1, 1.7) | 0.3 (-0.1, 0.8) | 0.5 (0.04, 1.0) | 0.6 (0.04, 1.2) | 0.3 (-0.5, 1.0) | 45.5 | 42.9 | 96.7 |
| **Mobility (% with any limitation)** | 3 | -14.2% | -14.2% | -14.1% | -5.8% | -7.2% | -3.2% | 40.8 | 50.7 | 22.9 |
| | 6 | -11.6% | -11.7% | -11.2% | -4.9% | -7.3% | 3.0% | 42.2 | 60.8 | 0 |
| **Social Isolation (score, 0–15)** | 3 | 0.4 (0.1, 0.7) | 0.6 (0.2, 0.9) | -0.09 (-0.6, 0.5) | 0.7 (0.4, 1.0) | 0.8 (0.5, 1.1) | 0.5 (0.1, 1.0) | 175.0 | 128.3 | 530.0 |
| | 6 | 0.2 (-0.01, 0.5) | 0.4 (0.05, 0.8) | -0.2 (-0.7, 0.4) | 0.4 (0.2, 0.7) | 0.5 (0.1, 0.8) | 0.3 (-0.1, 0.8) | 200.0 | 117.5 | 165.0 |
| **Loneliness (score, 3–9)** | 3 | -0.4 (-0.6, -0.3) | -0.4 (-0.6, -0.3) | -0.5 (-0.7, -0.2) | -0.2 (-0.3, -0.1) | -0.2 (-0.4, -0.08) | -0.1 (-0.3, 0.09) | 50.0 | 55.0 | 22.0 |
| | 6 | -0.3 (-0.5, -0.2) | -0.3 (-0.5, -0.2) | -0.4 (-0.6, -0.07) | -0.2 (-0.4, -0.1) | -0.3 (-0.4, -0.1) | -0.2 (-0.4, 0.04) | 66.7 | 90.0 | 47.5 |

d/wk: days/week; ND: no difference; Social Isolation: positive effect size indicates the intervention reduced social isolation; Loneliness: negative effect size indicates the intervention reduced feelings of loneliness.

[a] Effect size calculated as change from baseline as per McCrabb et al.[13]

months as compared with baseline (-0.3; 95% CI: -0.4, -0.1). Among older participants, loneliness score did not change significantly between baseline and 3 or 6 months.

At the 18-month follow-up, younger participants' loneliness scores were significantly higher than at the end of the intervention (+0.4; 95% CI: 0.2, 0.6). This resulted in no difference between loneliness scores at baseline and 18 months. Among older participants, loneliness scores also increased between 6- and 18-months (+0.6; 95% CI: 0.4, 0.9), which led to higher values at 18 months as compared with baseline (+0.4; 95% CI: 0.1, 0.7).

*Health-related quality of life*: Health-related quality of life as measured using the EQ-5D-5L index score did not change significantly over time in either age group. Among younger participants VAS score increased between baseline and 3 months (+6.9; 95% CI, 3.2 to 10.5), 6 months (+7.1; 95% CI, 3.3 to 10.8) and 18 months (+5.2; 95% CI, 1.3, 9.1). VAS score did not change significantly in older participants.

*Exploratory analysis of Winter 2020 cohort*: Among all participants in cycle 8 (Winter 2020), physical activity and mobility did not change significantly over time. However, social isolation scores decreased (indicating increased feelings of social isolation) between baseline and 3 months in younger (-1.8; 95% CI: -2.6, -1.0) and older participants (-2.5; 95% CI: -3.5, -1.4). Social isolation scores were also lower at the end of the intervention in younger (-1.4; 95% CI: -2.2, -0.5) and older (-1.7; 95% CI: -2.8, -0.6) participants. Although loneliness scores increased in the whole Winter 2020 cohort after 3 months (+0.3; 95% CI: 0.01, 0.6); change in loneliness score after 3 months was not significant in each age group. EQ-5D-5L and VAS scores did not change significantly over time in either age group.

## Voltage drop–objective 4

We compared the effect sizes between CTM Phases 1–2 and Phase 3 (Table 4). Among younger participants, the magnitude of the intervention effect dropped in Phase 3 compared with Phases 1–2 for physical activity, mobility and loneliness. However, CTM Phase 3 had a greater

positive effect on younger participants' social isolation than did the smaller scale Phases 1–2 intervention.

Among older participants, the positive impact of CTM on physical activity at 3 months was similar in magnitude between Phase 3 and Phases 1–2. However, we observed a voltage drop for both loneliness and mobility among participants. As per younger participants, after 3 months, the magnitude of the positive impact of CTM Phase 3 on older participants' social isolation was greater compared with Phases 1–2.

## Discussion

As public health researchers, we seek to narrow the "know-do gap" [53] and identify ways to deliver health-promoting interventions at broad scale. Here, we report the effectiveness of CTM Phase 3—a health-promoting intervention delivered at scale across the province of BC. We know of only two other scaled-up health-promoting interventions in older adults [18, 54]. Our study uses implementation science principles to fill knowledge gaps related to; i. CTM implementation and impact on participant-level outcomes after adapting the intervention to achieve 'best fit' [55] at scale-up, and ii. assessing the magnitude of 'voltage drop' after scale-up. Our hybrid effectiveness-implementation study of CTM Phase 3 builds on our previous reports of the design and community partnership development [19], formative evaluation [56], implementation of CTM at small scale [20, 21] and model adaptation [12]. Thus, this is not an isolated study—the innovation and importance of CTM Phase 3 can be better understood in the context of our earlier work.

### *Fidelity to intervention delivery was maintained following CTM program adaptations in Phase 3*

As per central tenets of successful implementation [28, 29], we engaged our key partners during pre-implementation planning, to co-design CTM as a flexible, scalable intervention [19], and to identify appropriate and feasible implementation strategies. For Phase 3 scale-up, we adopted a systematic process [57–59] to adapt CTM to meet the needs of older adults who lived in different geographic settings [12]. Such adaptations are often deemed necessary, and potentially inevitable to ensure that interventions are appropriate for new contexts and settings [16] and that scale-up is successful [60]. An opposing view suggests that adaptation is an implementation *failure* [29, 61]. The fine balance between adaptation and fidelity has been termed 'a dynamic tension' [60]—it reflects a potential chasm between what community delivery partners are willing and able to do, and researchers' goal to deliver the intervention as planned. The need to maintain this balance becomes more critical and more challenging as innovations are implemented at larger and larger scale.

In this third phase of CTM, we focused on adapting program content while retaining fidelity to core intervention components (one-on-one consultation, group meetings, check-ins). Core components are fundamental aspects of the intervention, considered potential drivers of positive health impact [62]. Changing core components risks altering aspects of intervention that are *perceived to* promote improved participant-level health outcomes. In our study, adaptation did not compromise overall fidelity of the CTM Phase 3 program as Activity Coaches delivered all CTM core components.

### *CTM when adapted and delivered at scale (Phase 3) led to short-term increases in older adults' physical activity*

CTM Phase 3 resulted in a modest increase in physical activity in all participants at 3 months (+1.0 days/week in younger participants; +0.9 days/week in older participants) that may lead to significant health benefits [63]. Whereas physical activity levels returned to

baseline levels among older participants during months 4–6 of the intervention, physical activity remained stable (did not drop) in younger participants during this period despite the lower program delivery 'dose' (i.e., no group meetings and only 3 telephone check-ins). Intervention-related benefits in physical activity were also maintained in younger participants 12-months after the intervention ended. This is a notable finding for two reasons. First, among older adults aged 74 years and younger, participation in CTM countered the trend that older adults' physical activity declines, on average, with age [64, 65]. Second, we conducted the latter part of our study during the turbulence of the COVID-19 pandemic when public health measures, such as stay at home directives, generally impacted older adults' daily routines. For 70.0% of participants, follow-up measurements were conducted during the first and second waves of the pandemic in BC. CTM group sessions were designed to instill long term skills such as goal setting and action planning; these lessons may have helped younger participants to maintain their physical activity during COVID. Although physical activity returned to baseline levels in older participants (≥75 years), it is also reassuring that physical activity did not decline further, particularly given the context of the COVID-19 pandemic. Additional program adaptation is likely required in order to maintain physical activity benefits among participants aged 75 years and older. Currently, we are exploring the efficacy of an extended contact intervention [66] (i.e., monthly or quarterly group meetings following completion of the main CTM intervention) on participant outcomes (NCT04592614).

Among younger participants, improved health outcomes were accompanied by increases in perceived health status—benefits that persisted at the 12-month follow-up. The perceived benefits in health status were within the range of values reported for the minimally important difference (MID) for the VAS. MID is defined as the smallest change in a patient-reported outcome that is perceived by patients as beneficial or that would result in a change in treatment [67] in clinical populations [68, 69]. However, we did not observe an improvement in the EQ-5D-5L index score in younger (or older) participants. Although the EQ-5D-5L demonstrated excellent psychometric properties (i.e., reliability and validity), this instrument may not be sensitive enough to monitor changes over time after participation in a health-promoting intervention such as CTM [70].

## Voltage drop is not inevitable–but can reflect program adaptation

Activity coaches and older adults who participated in CTM Phases 1–2, told us that social connectedness was central to program success, and encouraged us to place greater emphasis on opportunities for older adults to connect [12]. Therefore, in the CTM Phase 3 intervention we added a group meeting to promote social interactions among older adults. We also emphasized and formally integrated social connectedness into the activity coach training curriculum (e.g., Activity coaches offered more opportunities for group and pair discussion at group meetings).

Increased opportunities for CTM participants to connect with their peers in group meetings was reflected in diminished feelings of social isolation in Phase 3 as compared with Phases 1–2. What we observed was underscored by the positive association between program dose received (as reported by coaches) and decreased social isolation among older participants. Thus, it seems that voltage drop is not inevitable with program adaptations designed to improve outcomes.

However, unlike CTM Phases 1–2 [27], in CTM Phase 3 diminished social isolation following the intervention was not maintained at 12-month follow-up in younger or older participants. This is not surprising, as participants in four of seven CTM Phase 3 program cycles completed their follow-up during the COVID-19 pandemic. COVID public health 'stay-at-home' directives generally impacted older adults' ability to connect [71, 72]. Strategies to counter the harmful effects of social

isolation and loneliness on older adults' physical and mental health [73], might include effective health-promoting programs delivered at broad scale, such as CTM.

Adaptations to CTM Phase 3 may have reduced intervention-related benefits in physical activity (in younger participants only), mobility and loneliness when compared with CTM Phases 1 and 2. The magnitude of the voltage drop (median of 52.6%) for health outcomes at broad scale-up in CTM Phase 3 was similar to reports in a recent systematic review; across 10 eligible scaled-up physical activity interventions (targeting children, adults or older adults) the median voltage drop for physical activity was 59% [14].

Three factors may have contributed to voltage drop in physical activity, mobility and loneliness. First, some Phase 3 participants were older adults living in more remote areas of northern British Columbia where climate and limitations of the built environment contribute to decreased leisure-time activity options and limited access to active transportation, compared with more urban southern communities [74, 75]. Second, CTM Phase 3 participants had 4 fewer check-ins with their activity coaches. Check-in participation as reported by coaches was high in all CTM Phases, but slightly lower among Phase 3 participants (82.3% attended 70% of check-ins), as compared with Phases 1–2 participants (96.2% attended 70% of check-ins). Coaches were primary motivators for participants to 'stick with' their activity plans; reduced contact in Phase 3 may have contributed to lower check-in participation and ultimately to voltage drop in intervention effectiveness.

Third, Phase 3 activity coaches received their CTM training in a self-directed online format, whereas Phases 1–2 coaches were trained in person. Although most activity coaches (93.5%) found the online training useful, 19.6% of activity coaches were not fully confident in their ability to use what they learned in training. The CTM central support unit [32] provided support to activity coaches throughout the program. In future, it may be beneficial to offer activity coaches options to train either in-person or be self-directed, and between online group training sessions in person or on platforms like Zoom™.

## Limitations

We acknowledge several limitations. First, our cohort was comprised of mostly female older adults, most of whom self-identified as being white. This limits the external validity of our findings, and highlights the need for continued efforts to adapt health-promoting programs to reach older men, older adults from more diverse cultural and ethnic backgrounds, and people from marginalized groups. For example, to ensure a transgender-inclusive approach in future, we urge researchers to utilize a multidimensional measure to assess gender identity and lived gender [76] in addition to biological sex. Second, despite reasonable retention in Phase 3, those with better perceived health status were more likely to complete the program. The reasons for this must be more fully explored. Third, we did not assess whether intervention benefits were maintained beyond the 12-month follow-up. It will be important in future studies to assess if benefits can be maintained over a longer period of time. Finally, COVID-19 posed challenges for program delivery in the final cycle of Phase 3; the pandemic also affected participant follow-up for those who completed CTM prior to the pandemic. Although program delivery continued in cycle 8, 73.1% of programs pivoted to online delivery for at least one group meeting and 19.2% delivered more than half of group meetings online. Program attendance was similar to that observed in previous cycles; however, in our exploratory subgroup analysis of cycle 8 participants, we found, not surprisingly, that only social isolation scores changed during the 6-month intervention. Both younger and older participants reported feeling increasingly socially isolated whereas we did not observe any benefits of CTM on physical activity, mobility, loneliness or health-related quality of life in this cohort.

## Conclusions

Our findings highlight how continued program adaptation and refinement through a Knowledge to Action Cycle [77] are integral parts of the life-cycle of an intervention. Adaptations that accompany scale-up of evidence-based interventions (e.g., degree of fidelity/adaptation required to retain effectiveness), may help to preserve benefits (diminish voltage drop) as intervention are implemented across the scale-up continuum [8]. It seems important, in future, to quantify voltage drop in effective health-promoting interventions over the longer term and as they are implemented at increasing larger scale. Assessing the *scale-up penalty* may provide a more realistic assessment of intervention benefits at the population level. This may, in turn, reflect scalability of an intervention [8], and guide policy makers toward best investments in health promotion and public health [13]. We continue to advocate for hybrid effectiveness studies as both implementation and impact outcomes help us to understand why interventions do or do not 'work', especially at scale-up.

The public health and health promotion landscape may be forever changed, by the COVID-19 pandemic. There is a need for virtual health-promoting programs that can be adapted and delivered effectively, in the home environment. Flexible health-promoting programs–such as CTM—might be adapted (and evaluated) with community partners to meet the needs of more racially, culturally and gender diverse populations of older adults, who live in diverse settings.

## Supporting information

**S1 Table. Comparison of descriptive characteristics between participants who completed Choose to Move, those lost to follow-up and those in Cycle 8.** Baseline socio-demographic characteristics of participants who completed the Choose to Move intervention, those lost to follow-up and those in Cycle 8 (Winter 2020 cohort, excluded from the primary analysis). Values are n (%) or mean (standard deviation, SD).
(DOCX)

**S2 Table. Results of linear mixed model for impact outcome measures for Choose to Move participants in Cycle 8 (Winter 2020 cohort).** Adjusted means (95% confidence interval) for impact outcome measures by time point and age group in Choose to Move Phase 3 –Cycle 8 (Winter 2020 cohort; excluded from primary analysis due to impact of COVID-19 on program delivery).
(DOCX)

## Acknowledgments

We are grateful to the BC Ministry of Health and the Active Aging Society for their commitment to, and ongoing support of, Choose to Move. We also thank our delivery partner organizations, facility managers and coordinators, activity coaches, and all of the older adults who participated in Choose to Move. We are also appreciative of staff and trainees from the Active Aging Research Team for their assistance with data collection.

## Author Contributions

**Conceptualization:** Heather A. McKay, Lindsay Nettlefold, Adrian Bauman, Karim M. Khan, Joanie Sims Gould.

**Data curation:** Heather M. Macdonald, Lindsay Nettlefold.

**Formal analysis:** Heather M. Macdonald, Katie Weatherson.

**Funding acquisition:** Heather A. McKay, Joanie Sims Gould.

**Investigation:** Heather A. McKay, Joanie Sims Gould.

**Methodology:** Lindsay Nettlefold, Samantha M. Gray, Adrian Bauman, Joanie Sims Gould.

**Project administration:** Heather A. McKay, Lindsay Nettlefold, Joanie Sims Gould.

**Supervision:** Heather A. McKay, Joanie Sims Gould.

**Writing – original draft:** Heather A. McKay, Heather M. Macdonald, Katie Weatherson.

**Writing – review & editing:** Heather A. McKay, Heather M. Macdonald, Lindsay Nettlefold, Katie Weatherson, Samantha M. Gray, Adrian Bauman, Karim M. Khan, Joanie Sims Gould.

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
