## [Decision Letter · Decision Letter 0]

21 Nov 2022

PONE-D-22-11801What is the ‘voltage drop’ when an effective health promoting intervention for older adults—Choose to Move (Phase 3)—is implemented at broad scale?PLOS ONE

Dear Dr. Macdonald,

Thank you for submitting your manuscript to PLOS ONE. After careful consideration, we feel that it has merit but does not fully meet PLOS ONE’s publication criteria as it currently stands. Therefore, we invite you to submit a revised version of the manuscript that addresses the points raised during the review process.

Specifically, both reviewers raised concerns about the study. Please address all the comments point-by-point.

We look forward to receiving your revised manuscript.

Kind regards,

Jianhong Zhou

Staff Editor

PLOS ONE

Journal Requirements:

2. Please update your submission to use the PLOS LaTeX template. The template and more information on our requirements for LaTeX submissions can be found at http://journals.plos.org/plosone/s/latex

Reviewers' comments:

Reviewer's Responses to Questions

**Comments to the Author**

1. Is the manuscript technically sound, and do the data support the conclusions?

Reviewer #1: Yes

Reviewer #2: Partly

2. Has the statistical analysis been performed appropriately and rigorously? 

Reviewer #1: I Don't Know

Reviewer #2: Yes

3. Have the authors made all data underlying the findings in their manuscript fully available?

Reviewer #1: No

Reviewer #2: Yes

4. Is the manuscript presented in an intelligible fashion and written in standard English?

Reviewer #1: Yes

Reviewer #2: Yes

5. Review Comments to the Author

Reviewer #1: The paper ‘What is the ‘voltage drop’ when an effective health promoting intervention for older adults—Choose to Move (Phase 3)—is implemented at broad scale?’ discusses an important study on the scale up of a health promoting intervention. The paper is important as this wider implementation is often not discussed after smaller scale implementations.

Overall, the study seems like it was well planned and though through with suitable measures and data collection in place. I want to commend the authors on continuing this work through the pandemic, it was not an easy time. That said there are some instances where they have excluded data collected during the pandemic from analysis. I would be interested in this data, to see how the pandemic waves impacted on the fidelity and deliverables of this intervention, even if done as a separate analysis. This may have also added strength to the discussion where they discuss the impact of the ‘stay home’ mandates on the later cycles follow up and maintenance.

One of the main overarching issues for me that requires addressing is the framing of the study results. In the discussion broad statements such as ‘CTM when adapted and delivered at scale (Phase 3) increased older adults’ physical activity’ appeared to be misleading and unsupported by the results which showed much more nuance. For adults >75 years an increase was shown at 3 months, but not even maintained to 6 months. Further when the full sample is looked at, there is only a significant increase at 3 months, no significant difference from baseline at 6 and 18 months. I am disappointed that this nuance was not explored more. I see it as a strength to the paper that they are showing such interesting variations by age, however, critique of this was disappointingly absent (apart from in discussion with pandemic impact). We must be more critical and curious when we have an intervention that appears to work well for 3 months, but then work sporadically. Could this be when there is the change from small groups to check-ins? These interesting questions are not explored.

I have provided my main comments on the attached PDF document, and hope that they help the authors improve this paper, as it is a great study.

Reviewer #2: The study aims to assess the implementation, impact on physical activity, mobility, social isolation, loneliness, and health-related quality of life (impact outcomes), whether intervention effects were maintained and to compare voltage drop with previous CTM phases.

Abstract

Line 36, the word mean sd to be added.

Method

Line 133, the total duration of the program is to be stated.

Line 206, at least one decimal point for the percentage figures. This applies throughout the manuscript including tables where applicable.

The following information are to be added.

i) Line 218, the key name of the single-item physical activity

ii) Line 225, the name of the questionnaire.

iii) Line 247, one or two-tailed to be stated.

iv) Line 253, to be written as Fisher’s exact test and one or two-tailed test to be stated.

Line 222. ‘as no/any difficulty walking 400m’ to be rephrased.

Line 236, the word dimension or levels to be added to 3125.

Line 282, the sentence requires revision.

Results

Table 2, statistical tests to be denoted in the table footnote. However, based on CONSORT statement, all statistical tests for baseline comparison are to be avoided.

Table 3 requires cosmetic changes. The p values were not placed on the same row with the outcome, time point and age group. n for Mobility apart from % to be added and Bonferroni correction is to be denoted.

Tables 3 & 4, the adjusted variables and statistical tests to be denoted in the table footnote.

Table 4, 95%CI is to be stated in the title. ND is to be denoted in the table footnote.

Supplemental Table 1, statistical test to be denoted in the table footnote.

Supplemental Table 1, the table title ‘those in Cycle 8 (excluded from the analysis’) looks confusing. The comparison groups are to be clearly stated. For *significantly different from the Lost to Follow-up (p=0.033) and Cycle 8 (p=0.037) groups, the statistical test is to be stated.

Model fit to be discussed.

Figure 2, the outcome measures, activity coaches and participants engagement survey could be incorporated into the figure. Provincial and Comprehensive are to be clearly denoted in the Figure 2 footnote.

6. PLOS authors have the option to publish the peer review history of their article (what does this mean?). If published, this will include your full peer review and any attached files.

Reviewer #1: No

Reviewer #2: No

---

## [Author Response · Author response to Decision Letter 0]

10 Feb 2023

Reviewer #1: The paper ‘What is the ‘voltage drop’ when an effective health promoting intervention for older adults—Choose to Move (Phase 3)—is implemented at broad scale?’ discusses an important study on the scale up of a health promoting intervention. The paper is important as this wider implementation is often not discussed after smaller scale implementations.

Overall, the study seems like it was well planned and thought through with suitable measures and data collection in place. I want to commend the authors on continuing this work through the pandemic, it was not an easy time. That said there are some instances where they have excluded data collected during the pandemic from analysis. I would be interested in this data, to see how the pandemic waves impacted on the fidelity and deliverables of this intervention, even if done as a separate analysis. This may have also added strength to the discussion where they discuss the impact of the ‘stay home’ mandates on the later cycles follow up and maintenance.

One of the main overarching issues for me that requires addressing is the framing of the study results. In the discussion broad statements such as ‘CTM when adapted and delivered at scale (Phase 3) increased older adults’ physical activity’ appeared to be misleading and unsupported by the results which showed much more nuance. For adults >75 years an increase was shown at 3 months, but not even maintained to 6 months. Further when the full sample is looked at, there is only a significant increase at 3 months, no significant difference from baseline at 6 and 18 months. I am disappointed that this nuance was not explored more. I see it as a strength to the paper that they are showing such interesting variations by age, however, critique of this was disappointingly absent (apart from in discussion with pandemic impact). We must be more critical and curious when we have an intervention that appears to work well for 3 months, but then work sporadically. Could this be when there is the change from small groups to check-ins? These interesting questions are not explored.

I have provided my main comments on the attached PDF document, and hope that they help the authors improve this paper, as it is a great study.

Response: We thank the Reviewer for their positive and constructive feedback on our manuscript. Regarding the Reviewer’s first main comment, we appreciate the opportunity to clarify how the COVID-19 pandemic impacted fidelity of CTM and the deliverables of this intervention. Of the 26 programs in the Winter 2020 cycle, 20 programs modified mode of delivery (from in person to online) for at least one of the 5 group meetings; 5 of these programs delivered more than 50% of their group meetings online. Although attendance at the online group meetings was similar to group meeting attendance in the other seven cycles, physical activity opportunities in the community that were part of individual action plans were no longer available due to public health restrictions and physical distancing measures. Given the impact of COVID-19 on this cohort, we felt it was inappropriate to include them in our primary analysis. However, we agree with the Reviewer that it is important for us to share the data for this group. We now include a supplementary table with the results of the linear mixed effects model for the impact outcome measures in the Winter 2020 cohort. Please note that duration of follow-up was three months shorter for this group; we provide outcomes at baseline, 3 and 6 months. 

Regarding the Reviewer’s second comment, we did not intend to overstate the significance of our findings. We agree that the manuscript requires a more nuanced discussion of the apparent age-specificity of CTM’s benefit. In the Discussion, we now clarify the age-specificity of CTM’s impact on physical activity, noting that although CTM increased physical activity in all participants after 3 months, this benefit was only maintained in the younger group (<74 years) at 6 and 18 months. We agree that the lower program ‘dose’ during months 3-6 of the intervention may have contributed to the decline in physical activity among older participants. In the Discussion (Line 490), we now mention our current study that is investigating the efficacy of an extended contact intervention (i.e., monthly or quarterly group meetings following completing of the main CTM intervention). Continued contact with an activity coach may encourage participants, particularly those aged 75 years and older, to maintain their physical activity behaviours. 

1. Introduction, Line 62 & 70: Include page numbers with direct quotes

Action: We added the end quotes and the page number for the direct quote.

2. Introduction, Line 76: Do smaller effect sizes indicate voltage drop? 

Response: Yes, a smaller effect size in the scaled-up trial as compared with the pre-scale efficacy trial indicates voltage drop. 

3. Introduction, Line 94: Was the previous CTM study split to younger and older age group as in the present study? Provide examples of health benefits maintained 12 months after the intervention ended. 

Response: Yes, in the analysis of CTM Phases 1 and 2, we split the sample into the same age groups (60-74 yrs and > 75 yrs). Regarding maintenance of health benefits 12 months after CTM ended, we found that in the younger group, participants maintained their intervention benefits in mobility, social isolation and loneliness. Among older participants, only intervention-related decreases in loneliness were maintained 12 months after the intervention. We added this information to the introduction.

Action: Page 5, Lines 95-99 now read: Some health benefits were maintained 12 months after the intervention ended [27]. Specifically, among younger participants (60-74 years), intervention-related benefits in mobility, social isolation and loneliness were maintained 12 months after CTM ended, whereas in older participants (> 75 years), only decreased loneliness was maintained over 12 months.

4. Introduction, Line 98: Was there a numbers goal of how many locations/people involved? 

Response: The number of programs we aimed to deliver in CTM Phase 3 (n=135) was determined according to our commitment to the British Columbia Ministry of Health (who provided funds to support delivery of CTM). With an expected average of 12 participants per program, we anticipated a total of 1620 older adult participants. Ultimately, we delivered more programs (n=165) because of a lower number of participants per program. 

5. Methods: Line 115: Unnecessary to include the full definition of an observational study, be concise in how your study aligned with the definition. 

Response: We removed the definition of an observational study. 

6. Methods, Line 171: Move description of number of participants with consent to the Results.

Response: Thank you for this suggestion. We moved this information to the opening paragraph of the Results. 

7. Methods, Line 175: (regarding exclusion of the Winter 2020 cycle) This seems really unfortunate to not include this subgroup in some form of analysis. If this is not going to be published elsewhere in the future, I think it should be included as a subgroup.

Response: Thank you for this suggestion. We now include the analysis of the Winter 2020 cohort in Supplementary Table 2 and summarize the results for this exploratory subgroup analysis. We also provide a brief interpretation of these findings in the Discussion.

Action: Lines 417-25 now read: Among all participants in cycle 8 (Winter 2020), physical activity and mobility did not change significantly over time. However, social isolation scores decreased (indicating increased feelings of social isolation) between baseline and 3 months in younger (-1.8; 95% CI: -2.6, -1.0) and older participants (-2.5; 95% CI: -3.5, -1.4). Social isolation scores were also lower at the end of the intervention in younger (-1.4; 95% CI: -2.2, -0.5) and older (-1.7; 95% CI: -2.8, -0.6) participants. Although loneliness scores increased in the whole Winter 2020 cohort after 3 months (+0.3; 95% CI: 0.01, 0.6); change in loneliness score after 3 months was not significant in each age group. EQ-5D-5L and VAS scores did not change significantly over time in either age group.

Action: Lines 561-8 now read: Finally, COVID-19 posed challenges for program delivery in the final cycle of Phase 3; the pandemic also affected participant follow-up for those who completed CTM prior to the pandemic. Although program delivery continued in cycle 8, 76.9% of programs pivoted to online delivery for at least one group meeting and 19.2% of these programs delivered more than half of group meetings online. Program attendance was similar to that observed in previous cycles; however, in our exploratory subgroup analysis of cycle 8 participants, we found, not surprisingly, that only social isolation scores changed during the 6-month intervention. Both younger and older participants reported feeling increasingly socially isolated whereas we did not observe any benefits of CTM on physical activity, mobility, loneliness or health-related quality of life in this cohort.

8. Methods, Line 214: On the PDF, the Reviewer added the comment “Why?” next to the ethnicity categories, which we assume to mean that they would like us to clarify why we collapsed information on ethnicity into three categories. 

Response: We reduced the number of categories to three for inclusion in our statistical analysis. The three categories we chose (Asian, white or other/mixed) reflect the composition of our sample and align with Statistics Canada classifications (https://www12.statcan.gc.ca/census-recensement/2016/ref/guides/006/98-500-x2016006-eng.cfm). 

9. Table 2: Why are baseline demographics presented by age group and not by cycle? 

Response: We chose to present baseline demographics by age group rather than by program cycle because the variability in demographic outcomes was greater between age groups than between program cycles. 

10. Table 2: Error in number of participants by age group. 

Response: Thank you for noting this error. We corrected the number of participants by age group in Table 2. 

11. Table 2: Clarify why the n differs for some outcomes. 

Action: We added a footnote to the table to indicate that the n differs for some outcomes because of missing data. 

12. Results, Line 330: Why was participant responsiveness only looked at after the first 3 months?

Response: Thank you for the opportunity to clarify our measure of participant responsiveness. Questions related to participant responsiveness were only asked after the first 3 months of the intervention because coaches were asked to reflect on participants’ level of engagement at the group meetings only. Coaches were not asked to comment on participants’ engagement during the telephone check-ins that took place throughout the 6-month intervention. We clarified in this section that group meetings only took place during the first 3 months of the intervention. 

Action: Line 349-50 now reads: After the first 3 months of the intervention (period during which all group meetings were held), 90.4% of participants were satisfied with CTM.

13. Discussion, Line 427: This feels misleading. There is only evidence that it leads to an increased level of activity in younger adults. For older adults an increased was shown at three months, but not even maintained to 6 months, in your own words 'For older participants, physical activity at 18 months did not differ from values at 6 months, or at baseline.' Further when the full sample is looked at, there is only a significant increase at 3 months, no significant difference from baseline at 6 and 18 months.

Response: We agree with the Reviewer that this subheading may be considered misleading in the context of the full 6-month intervention, and the apparent age-specificity of intervention effectiveness. We modified the subheading accordingly. 

Action: Lines 469-70 now read: CTM when adapted and delivered at scale (Phase 3) led to short-term increases in older adults’ physical activity

14. Discussion, Line 434: (in response to statement that CTM countered the trend that older adults’ PA declines with age) How? Activity levels were similar between older and younger adults at baseline of this study, and we saw differences in how they maintained physical activity. Additionally, the older adult group showed no change from baseline at 6 and 18 months.

Response: Thank you for the opportunity to clarify our thoughts on this point. Intervention benefits in physical activity were only maintained in the younger group in the year after the intervention ended. Thus, it is possible that in this age group CTM countered the trend that older adults’ physical activity declines with age. Similarly, although intervention benefits were not maintained in the older age group, it is reassuring that physical activity levels did not decrease below baseline values, particularly given the context of the COVID-19 pandemic. Additional program adaptation may be warranted to target maintenance behaviours in those aged 75 years and older. 

Action: Lines 476-492 now read: Intervention-related benefits in physical activity were also maintained in younger participants 12-months after the intervention ended. This is a notable finding for two reasons. First, among older adults aged 74 years and younger, participation in CTM countered the trend that older adults’ physical activity declines, on average, with age [63, 64]. Second, we conducted the latter part of our study during the turbulence of the COVID-19 pandemic when public health measures, such as stay at home directives, generally impacted older adults’ daily routines. For 70.0% of participants, follow-up measurements were conducted during the first and second waves of the pandemic in BC. CTM group sessions were designed to instill long term skills such as goal setting and action planning; these lessons may have helped younger participants to maintain their physical activity during COVID. Although physical activity returned to baseline levels in older participants (>75 years), it is also reassuring that physical activity did not decline further, particularly given the context of the COVID-19 pandemic. Additional program adaptation is likely required in order to maintain physical activity benefits among participants aged 75 years and older. Currently, we are exploring the efficacy of an extended contact intervention [65] (i.e., monthly or quarterly group meetings following completion of the main CTM intervention) on participant outcomes (NCT04592614).

15. Discussion, Lines 438-440: Again, this doesn't align with your results. Both the pooled and >75 adults did not maintain their physical activity levels post 3 months.

Response: Thank you for noting this inconsistency. We clarified that CTM group sessions may have helped younger participants to maintain their physical activity during the pandemic. We also now comment on the need to adapt CTM to better support participants aged 75 years and older. 

Action: Please see the Action for Comment 14 above. 

16. Discussion, Line 450: Would we expect an impact on EQ-5D-5L when we haven’t seen any lasting impact on loneliness, social isolation, mobility or activity?

Response: CTM positively impacted physical activity, mobility, social isolation and loneliness among younger participants, and thus, the increased score on the EQ-5D-5L Visual Analog Scale aligns with these results. The VAS is a more subjective rating of health whereas the EQ-5D-5L score is generated by converting a 5-digit health state profile (which represents self-reported abilities/problems on 5 dimensions of health) into an index score using a value set that reflects the preferences of the general population. Further, as noted by Feng et al. in their systematic review (2021), additional study is needed to determine responsiveness and sensitivity to change of the EQ-5D-5L, and define what a relevant change is for this tool. 

17. Discussion, Lines 471-472: These broad statements of CTM effectiveness again feel misleading.

Response: We did not intend to mislead readers with this statement. We acknowledge the wording of this sentence could be more specific; therefore we changed the text to make it more clear that we refer to adaptations to CTM in Phase 3 that may have led to the voltage drop we observed for some outcomes, as compared with changes we observed in Phases 1 and 2. 

Action: Lines 524-526 now read: Adaptations to CTM Phase 3 may have reduced intervention-related benefits in physical activity (in younger participants only), mobility and loneliness when compared with CTM Phases 1 and 2.

18. Discussion, Line 485: Any theories as to why lower in phase 3?

Response: Check-in participation may have been lower in Phase 3 as compared with Phases 1 and 2 because of the lower overall dose delivered in Phase 3. This may have resulted in less commitment from participants. Alternatively, it is possible that coaches may have been unable to reach some participants by phone. However, we did not ask coaches to document how many times they tried to reach participants during the intervention. 

Action: Lines 537-540 now read: Coaches were primary motivators for participants to ‘stick with’ their activity plans; reduced contact in Phase 3 may have contributed to lower check-in participation and ultimately to voltage drop in intervention effectiveness.

Reviewer #2: The study aims to assess the implementation, impact on physical activity, mobility, social isolation, loneliness, and health-related quality of life (impact outcomes), whether intervention effects were maintained and to compare voltage drop with previous CTM phases.

1. Abstract, Line 36, the word mean sd to be added.

Action: Line 36 now reads: We conducted a type 2 hybrid effectiveness-implementation pre-post study of CTM; older adult participants (n=1012; mean age 72.9, SD = 6.3 years; 81% female) were recruited by community delivery partners.

2. Methods, Line 133, the total duration of the program is to be stated.

Action: Line 133 now reads: During the first 3 months of the 6-month program, activity coaches provided participants…

3. Line 206, at least one decimal point for the percentage figures. This applies throughout the manuscript including tables where applicable.

Action: We added one decimal point to all percentage figures in the text and tables. 

4. The following information is to be added.

i) Line 218, the key name of the single-item physical activity. 

Response: As per the work of Milton et al., the name of the questionnaire is the single-item measure. 

Action: None

ii) Line 225, the name of the questionnaire.

Response: The 3-item social isolation questionnaire does not have a formal name. We adapted the original two questions from Vernoff et al., (ref 48) (How often do you visit with neighbours? How often do you get together with friends) to characterize social isolation as the weekly frequency of: 1) visiting with friends, neighbours and/or relatives; 2) phone/email exchanges with friends, neighbours and/or relatives; and 3) attending programs/groups/clubs/organizations. 

Action: Lines 233-34: We added the name of the loneliness questionnaire (UCLA Loneliness Scale, UCLA-3), but did not add a name for the social isolation questionnaire.

iii) Line 247, one or two-tailed to be stated.

Action: Lines 252-6 now read: With alpha=0.05 (one-tailed) and an estimated sample of 963 participants (1620*70% recruitment*15% attrition), we would have >95% power to detect a meaningful change in physical activity of 1 day/week between baseline and 6 months and >90% power to detect a change of 0.5 day/week.

iv) Line 253, to be written as Fisher’s exact test and one or two-tailed test to be stated.

Action: Lines 262-5 now read: We used two-tailed chi-squared or Fisher’s exact test for categorical variables (sex, age category, ethnicity, education, chronic conditions, mobility limitations, and subset participation) and analysis of variance for continuous variables (body mass index and impact variables).

5. Line 222. ‘as no/any difficulty walking 400m’ to be rephrased.

Action: Lines 227-9 now read: Participants also self-reported their capacity for mobility at each time point; we dichotomized responses as either NO difficulty or ANY difficulty walking 400m and/or climbing one flight of stairs [47].

6. Line 236, the word dimension or levels to be added to 3125.

Response: The EQ-5D-5L asks respondents to indicate their health state for each of 5 dimensions. Responses are coded as single-digit numbers (range 1-5) and then combined into a 5-digit code (e.g., 11245) to describe the overall health state. Therefore, there are 55 (=3125) possible health states. Based on this description, we don’t feel it is necessary to add the word dimension or levels to 3125 in this sentence. 

Action: None

7. Line 282, the sentence requires revision.

Action: Lines 295-7 now read: We then calculated the voltage drop (i.e., percent of the effect size reported in CTM Phases 1-2 that was retained in Phase 3) as: (Phase 3 effect size / Phases 1-2 effect size)*100 [20].

8. Results, Table 2, statistical tests to be denoted in the table footnote. However, based on CONSORT statement, all statistical tests for baseline comparison are to be avoided.

Action: We added information on statistical tests to the table footnote. We feel it is appropriate to conduct baseline comparisons because this was not a randomized trial. 

9. Table 3 requires cosmetic changes. The p values were not placed on the same row with the outcome, time point and age group. n for Mobility apart from % to be added and Bonferroni correction is to be denoted.

Response: Thank you for noting the missing details and formatting issue. While we appreciate the Reviewer’s feedback on the table format, we believe the original format is easiest to follow with respect to the time comparisons. For example, the first p-value corresponds to the 0-3 month comparison. If we placed this p-value on the first row, the reader may assume (without looking at the column heading) that the p-value is for baseline comparisons between age groups. We are happy to reconsider the Reviewer’s suggestion or another formatting suggestion from the Editorial Team if needed.

Action: Table 3- We added the n for Mobility and updated the percent values to include one decimal place. We also added information on the Bonferroni correction to the table footnote. 

10. Tables 3 & 4, the adjusted variables and statistical tests to be denoted in the table footnote.

Action: We added details on the statistical tests to the table footnote. 

11. Table 4, 95%CI is to be stated in the title. ND is to be denoted in the table footnote.

Action: We added 95% confidence interval to the title and defined ND in the table footnote.

12. Supplemental Table 1, statistical test to be denoted in the table footnote.

Action: We added information on the statistical tests to the table footnote. 

13. Supplemental Table 1, the table title ‘those in Cycle 8 (excluded from the analysis’) looks confusing. The comparison groups are to be clearly stated. For *significantly different from the Lost to Follow-up (p=0.033) and Cycle 8 (p=0.037) groups, the statistical test is to be stated.

Response: We agree that the wording of the table caption be modified to clarify that participants in Cycle 8 were excluded from our primary analysis. 

Action: Supplemental Table 1 heading now reads: Baseline socio-demographic characteristics of participants who completed the CTM intervention, those lost to follow-up and those in Cycle 8 (excluded from the primary analysis). Values are n (%) or mean (standard deviation, SD).

14. Model fit to be discussed.

Response: We assessed model fit graphically using residual plots. All plots indicated acceptable model fit. We now mention this in the methods.

Action: Lines 281-2 now reads: We assessed model ﬁt graphically using residual plots; plots indicated acceptable model fit.

15. Figure 2 (participant flow diagram), the outcome measures, activity coaches and participants engagement survey could be incorporated into the figure. Provincial and Comprehensive are to be clearly denoted in the Figure 2 footnote.

Response: Thank you for this suggestion. We added details on outcome measures, activity coaches and participant engagement survey to the figure, and denoted the different cohorts in the Figure 2 caption.

---

## [Decision Letter · Decision Letter 1]

20 Apr 2023

What is the ‘voltage drop’ when an effective health promoting intervention for older adults—Choose to Move (Phase 3)—is implemented at broad scale?

PONE-D-22-11801R1

Dear Dr. Macdonald,

We’re pleased to inform you that your manuscript has been judged scientifically suitable for publication and will be formally accepted for publication once it meets all outstanding technical requirements.

Kind regards,

Hugh Cowley

Staff Editor

PLOS ONE

Additional Editor Comments (optional):

Reviewers' comments:

Reviewer's Responses to Questions

**Comments to the Author**

1. If the authors have adequately addressed your comments raised in a previous round of review and you feel that this manuscript is now acceptable for publication, you may indicate that here to bypass the “Comments to the Author” section, enter your conflict of interest statement in the “Confidential to Editor” section, and submit your "Accept" recommendation.

Reviewer #1: All comments have been addressed

Reviewer #2: All comments have been addressed

2. Is the manuscript technically sound, and do the data support the conclusions?

Reviewer #1: Yes

Reviewer #2: Yes

3. Has the statistical analysis been performed appropriately and rigorously? 

Reviewer #1: Yes

Reviewer #2: Yes

4. Have the authors made all data underlying the findings in their manuscript fully available?

Reviewer #1: Yes

Reviewer #2: Yes

5. Is the manuscript presented in an intelligible fashion and written in standard English?

Reviewer #1: Yes

Reviewer #2: Yes

6. Review Comments to the Author

Reviewer #1: Thank you for you diligent and respectful response to my comments. I am really happy with the quality and presentation of the paper in its current format. I wish you all the best

Reviewer #2: (No Response)

7. PLOS authors have the option to publish the peer review history of their article (what does this mean?). If published, this will include your full peer review and any attached files.

Reviewer #1: **Yes: **Fay Manning

Reviewer #2: No

---

## [Editor Report · Acceptance letter]

27 Apr 2023

PONE-D-22-11801R1 

What is the ‘voltage drop’ when an effective health promoting intervention for older adults—Choose to Move (Phase 3)—is implemented at broad scale? 

Dear Dr. Macdonald:

I'm pleased to inform you that your manuscript has been deemed suitable for publication in PLOS ONE. Congratulations! Your manuscript is now with our production department. 

Kind regards, 

on behalf of

Mr Hugh Cowley 

Staff Editor

PLOS ONE